# Communication-Efficient Federated Learning via Model Update Distillation

## Abstract

Federated learning (FL) is a popular distributed machine learning framework for edge computing. However, it faces a significant challenge: the communication overhead caused by frequent model updates between clients and the central server. Previous studies have overlooked a crucial piece of information: the central server already knows the initial model on each client before local training begins in every round. This oversight leads to significant redundancy in communication, as full model information are transmitted unnecessarily. To address this, we propose a novel framework called *model update distillation* (MUD), which leverages this prior knowledge to decouple model parameters from the network architecture. Instead of transmitting raw parameter updates, our method synthesizes and transmits compact tensor sequences that encode only the essential information for synchronization. This dramatically reduces communication overhead while still allowing recipients to accurately reconstruct the intended model updates. Extensive experimental results demonstrate that FedMUD achieves substantial improvements in communication efficiency, making it a highly effective solution for federated learning in bandwidth-constrained environments. The PyTorch-like core code can be found in 3.

## 1 Introduction

Federated learning (FL) (McMahan et al., 2017; Shokri & Shmatikov, 2015) has become a promising approach for privacy-preserving machine learning by allowing model training directly on edge devices, such as smartphones and IoT sensors, without requiring centralized data storage. This decentralized framework ensures that sensitive data remains on-device, mitigating privacy and regulatory concerns (Ching et al., 2018; GDPR, 2016; ADPPA, 2022). With the rise of edge computing, FL has gained further traction, enabling real-time data processing and decision-making at the source (Feng et al., 2021; Nguyen et al., 2021). By leveraging the growing computational power of mobile devices, FL not only enhances scalability and efficiency in diverse fields like healthcare, finance, and smart cities but also reduces latency and the need for transmitting raw data to centralized servers.

However, one of the persistent challenges in FL is the growing imbalance between computing power and communication bandwidth. As shown in Table 1, advances in bandwidth have not kept pace with computing power. For instance, MediaTek and Qualcomm chips have improved by 53% and 31%, respectively, over their previous generations (PrimateLabs, 2024). In contrast, the global median upload bandwidth for mobile devices increased by only 7% from 2023 to 2024 (Ookla, 2024). This growing disparity highlights a key challenge in FL: modern neural network models, even lightweight ones like MobileNet, still consist of millions of parameters. Given that edge devices typically rely on wireless or long-distance connections, with bandwidth often limited to tens of Mbps, this bandwidth bottleneck severely restricts FL's potential. As computational power continues to rise, optimizing communication efficiency in bandwidth-constrained environments is crucial for maintaining FL performance.

To address this challenge, communication-efficient FL has emerged as an active research area. Researchers have developed various strategies to reduce the communication burden, including techniques like model compression (e.g., quantization (Liu et al., 2023; Sun et al., 2022) and sparsification (Aji & Heafield, 2017; Dai et al., 2022)) that minimize the size of model updates. Additionally, methods like delayed gradient averaging allow devices to perform more local computation before transmitting

Table 1: Comparison of Median Mobile Upload Bandwidth vs. Chip Computing Power Growth.

| Mobile Bandwidth | | MediaTek | | Qualcomm | |
|---|---|---|---|---|---|
| Year | Upload (Mbps) | Processer | Geekbench 6 | Processer | Geekbench 6 |
| 2023 | 10.26 | Dimensity 9200 | 5119 | Snapdragon 8 Gen 2 | 5697 |
| 2024 | **11.02 (+7%)** | Dimensity 9300 | **7857 (+53%)** | Snapdragon 8 Gen 3 | **7466 (+31%)** |

updates, thus reducing communication frequency (Zhu et al., 2021). Despite these advancements, the reduction in communication overhead remains limited due to the fundamental relationship between number of model parameters and network architecture. This interdependence means even minor updates of each parameter may require transmitting full number of parameters, leading to continued communication inefficiencies.

In this paper, we uncover a critical yet often overlooked information in federated learning: in each training round, **the central server has prior knowledge of the initial model** that each edge client will use for local training. By exploiting this valuable information, we propose a novel federated learning framework called Model Update Distillation (FedMUD). Inspired by gradient inversion (Zhu et al., 2019) and dataset distillation (Wang et al., 2018), FedMUD constructs a synthetic tensor sequence, which is fed into the initial model of the current round. Specially, after local update, we fixed the initial model parameters and iteratively optimize this tensor sequence to ensure that the resulting parameter differences closely match the actual local updates produced by the clients. Through model update distillation, we condense these parameter differences into a compact tensor sequence, allowing recipients to accurately reconstruct the intended model updates from the sender. This decoupling of model updates from the full parameter set significantly reduces communication overhead, as it eliminates the need to transmit the entire array of raw parameters. By treating the model's parameter updates as a whole and compressing them, our approach offers a promising solution to improve communication efficiency.

The main contributions of this paper are summarized as follows:

- We point out and exploit a crucial piece of information that has been overlooked in previous research: the central server in federated learning knows the initial model on each client before local training begins in every round.
- We propose FedMUD, a new framework that treats network parameter updates as a whole, decoupling them from the network architecture and ensuring a more efficient learning process.
- Experimental results demonstrate that model update distillation significantly reduces the amount of communication compared to baselines, without significant accuracy degradation, enabling a communication-efficient and training-accelerated FL process.

## 2 RELATED WORKS

**Communication Frequency Reduction.** This strategy involves allowing devices to perform multiple local updates before transmitting their model updates to the central server, thereby reducing the frequency of communication rounds (McMahan et al., 2017; Haddadpour et al., 2019; Zhu et al., 2021). While this approach reduces the overall transmission data amount, it may result in slower convergence and potential overfitting if not carefully managed.

**Gradient Information Compression.** This strategy focuses on compressing gradients (Liu et al., 2023; Reisizadeh et al., 2020). This includes methods such as quantization (Hönig et al., 2022), which reduces the precision of gradient values to decrease the transmitted information volume, and sparsification (Aji & Heafield, 2017; Dai et al., 2022), which involves sending only a subset of gradients by retaining the most significant ones and setting others to zero. However, aggressive quantization and sparsification can lead to information loss and potentially hinder model accuracy.

**Networks Pruning.** This strategy attempts to remove less influential weights or neurons from the model, effectively reducing the parameter count and thus the size of gradients (Zhu et al., 2022; Wang

(a) Overview of FedMUD

(b) FedAvg Pipeline

(c) FedMUD Pipeline

Figure 1: Overview of FedMUD and the pipelines of FedAvg and FedMUD. (a) illustrates a four-device scenario where two resource-constrained devices skip local distillation, while the other two perform full MUD, achieving bi-directional communication efficiency. (b) depicts the FedAvg pipeline, which incurs significant communication overhead. (c) shows the FedMUD pipeline, where model updates are compressed before transmission and reconstructed afterward, reducing communication time through additional computation.

et al., 2022). However, pruning-based methods require careful hyperparameter tuning and may result in model degradation if not applied judiciously.

**Compact Proxy Transmission.** This strategy focuses on uploading logits (Sattler et al., 2020; Shao et al., 2024) or dataset representations (Xiong et al., 2023; Castiglia et al., 2023) or proxy models (Kalra et al., 2023; Wu et al., 2022). Instead of transmitting raw gradients, devices calculate and send logits for their data samples to the central server. Alternatively, devices can convey aggregated representations of their datasets, such as centroids or other statistical summaries (Liu et al., 2022), which serve as a compact proxy for the raw gradients, and potential impact on model performance.

## 3 MODEL UPDATE DISTILLATION

As illustrated in Fig. 1, in this paper, we explore FL across $N$ edge devices, each characterized by varying computational and bandwidth resources. Consider an edge device $i$, which holds a private local dataset $\mathcal{D}_i = \{(\boldsymbol{x}_j^i, y_j^i)\}_{j=1}^{m_i}$, where the data points are sampled from a distinct distribution $\mathcal{P}_i$ over the space $\mathcal{X} \times \mathcal{Y}$. The central idea of FL is to collaboratively train a global model without directly sharing the private data. This is achieved through a process where edge devices intermittently send their local model updates to a central server, subsequently receiving an updated global model in return. The primary aim of FL is to develop a global model that effectively minimizes the combined

risk across all private datasets:

$$\arg\min_{\omega} \mathcal{L}(\omega, \mathcal{D}) \triangleq \frac{1}{N} \sum_{i=1}^{N} \mathcal{L}_i(\omega, \mathcal{D}_i), \tag{1}$$

where $\omega$ denotes the global model parameters. Here, $\mathcal{L}_i(\omega, \mathcal{D}_i) = \frac{1}{m_i} \sum_{j=1}^{m_i} \ell\left(\omega; \left(\boldsymbol{x}_j^i, y_j^i\right)\right)$ represents the empirical risk for device $i$, with $\ell$ being the loss function, and $\mathcal{D}$ symbolizing the overall training dataset. Our overarching goal is to efficiently reduce both the communication overhead and the total wall-clock time required for FL.

## 3.1 MOTIVATION

In the standard FL process, the central server distributes the aggregated global model from the previous round to each participating device. These devices then perform local training using the global model as a starting point, updating their local models accordingly. The FedAvg algorithm a cornerstone of FL, involves sending model updates between the new and old models back to the server. However, this approach has a significant inefficiency: the data volume transferred in each communication round remains constant, irrespective of the extent of changes in model parameters, as long as the network architecture does not change. This results in redundant transmission of update information. The root of this issue lies in the inherent interdependence between the model parameters and the network architecture in traditional FL approaches. Since the network's structure and its parameters are closely linked, even minor changes in parameters require retransmitting the entire model, leading to inefficiency in communication.

## 3.2 OUR STRATEGY

We introduce a novel approach that involves approximating model updates across different training periods within a single model. This approximation is represented as a synthetic tensor sequence, drawing inspiration from concepts in dataset distillation (Wang et al., 2018) and gradient inversion (Zhu et al., 2019). The core idea is for the sender to transmit this tensor sequence, which encapsulates only the indispensable information required for model updates, thus eliminating the need to transmit the entire set of raw parameter differences. Upon receiving the tensor sequence, the recipient can accurately reconstruct the original model update by performing a single gradient descent step, integrating the information from the received tensor sequence with the state of the previous model.

---

**Algorithm 1:** Model Update Distillation

**Input:** Tensor length $m$, model update $\Delta\omega(t_1, t_2) = \omega^{(t_1)} - \omega^{(t_2)}$, number of iterations $K$
**Initialization:** Initialize the tensor sequence $\zeta_0 = \{(\hat{x}_j, \hat{y}_j)\}_{j=1}^{m}$ with the same dimension as local samples
**for** $q = 0, \ldots, K-1$ **do**
  Derive synthetic gradient using using one-step SGD: $\Delta\omega_{syn} \leftarrow \nabla_{\omega} F_i(\omega, \zeta_q)$
  Calculate MSE loss: $\mathcal{L} = \text{MSE}(\Delta\omega(t_1, t_2), \Delta\omega_{syn})$
  Update tensor sequence: $\zeta_{q+1} = \text{optim}_{LBFGS}(\zeta_q, \mathcal{L})$
**end**
**Return:** $\zeta_K = \{(\hat{x}_j, \hat{y}_j)\}_{j=1}^{m}$

---

In the following, we detail the optimization mechanics of MUD. For a model with parameters $\omega$, we define its parameter difference between two periods $t_1$ and $t_2$ as $\Delta\omega(t_1, t_2) = \omega^{(t_1)} - \omega^{(t_2)}$. By approximating $\Delta\omega(t_1, t_2)$, we can update the model parameters at timestamp $t_2$ even when we only have access to the old model $\omega^{(t_1)}$. To achieve this, we synthesize a tensor sequence $\zeta = \{(\hat{x}_j, \hat{y}_j)\}_{j=1}^{m}$, which is tailored to approximate the parameter difference using one-step gradient descent on $\omega^{(t_1)}$. Our objective is to *discover the shortest projected path between $\omega^{(t_1)}$ and $\omega^{(t_2)}$*. To synthesize the sequence, we minimize the error between the parameter difference and the accumulated gradient of the sequence on $\omega^{(t_1)}$:

$$\zeta = \arg\min_{\{(x_j, y_j)\}_{j=1}^{m}} \left\| \Delta\omega(t_1, t_2) - \sum_{j=1}^{m} \frac{\partial \ell\left(\omega^{(t_1)}; (x_j, y_j)\right)}{\partial \omega^{(t_1)}} \right\|_2^2, \tag{2}$$

where $m$ represents the solved length of the optimal sample sequence. The recovery process for $\omega^{(t_2)}$ involves a one-step gradient descent on $\zeta$:

$$\omega^{(t_2)} = \omega^{(t_1)} - \sum_{(\hat{x},\hat{y}) \in \zeta} \frac{\partial \ell(\omega^{(t_1)}; (\hat{x}, \hat{y}))}{\partial \omega^{(t_1)}}. \tag{3}$$

The workflow for MUD is shown in Algorithm 1 and the source code is provided in Algorithm 3. Firstly, an initialization step is executed to create a tensor sequence, denoted as $\zeta_0$, which comprises $m$ synthetic samples. These samples are dimensionally equivalent to the local samples. Subsequent iterations are conducted to align the synthetic gradients with the actual updates of the local model, symbolized as $\Delta\omega$. Specifically, in the $q$-th iteration, based on the previously generated tensor sequence $\zeta_q$, a one-step stochastic gradient descent (SGD) is applied. This step aims to approximate the real difference in model parameters $\Delta\omega$. To quantify this difference, we introduce the Mean Squared Error (MSE) loss, which then serves as the objective for minimization, tackled using the LBFGS optimizer. Through this mechanism, we are able to update the tensor sequence to $\zeta_{q+1}$.

As the model size increases, the approximation error between the virtual gradient obtained by the MUD method and the real model parameter difference tends to grow, especially for large-scale neural networks. To address this, we introduce a modular alignment approach that segments the network into modules. Each module uses an independent synthetic dataset to approximate its gradient change. The number of modules is kept below the local training batch size to ensure it stays within the device's computational capacity. By modularizing the network, model update distillation for each sub-module can be processed in parallel, reducing delays. We will analyze the error introduced by MUD in the convergence analysis section.

### 3.3 Differences with Distillation-related Methodologies

We delve into the differences between the MUD and the popular distillation-related methodologies:

**– Knowledge Distillation (KD)** is a process involving two distinct neural network sets: a larger, complex "teacher" model, and a smaller, efficient "student" model (Hinton et al., 2015). The primary goal of KD is to transfer the knowledge from the teacher to the student, enabling the student model to reach performance levels similar to the teacher. This transfer is generally accomplished by minimizing the differences in predictions (probabilities or logits) between the teacher and student models when given the same input data.

**– Dataset Distillation (DD)** represents an extension of KD where knowledge transfer occurs at the dataset level rather than between models (Wang et al., 2018). The core idea behind DD is to condense a large and comprehensive training dataset into a much smaller, but highly representative, synthetic dataset, given a fixed network initialization. This smaller dataset is designed to retain the essential characteristics and information of the original dataset, enabling models to be trained effectively on this distilled dataset instead of the full, larger dataset.

**– Differences with KD and DD:** From the description provided, it is evident that our proposed MUD method diverges significantly from both KD and DD in its approach and objectives. Unlike KD, which is centered on the relationship between a larger, more complex "teacher" model and a smaller, more efficient "student" model, our MUD focuses on the update within a single model across different training periods. The primary input in MUD is the parameter disparity observed between two consecutive training stages of the same model. This approach is fundamentally different from the input used in DD, which involves a large-scale dataset that typically contains thousands to millions of images. The output of MUD contrasts sharply with the output of KD, which is the student model trained to mimic the teacher model. Although MUD's output bears a resemblance to DD in terms of its form, the two are fundamentally different in their design intentions.

## 4 Model Update Distillation based Communication-Efficient FL

We now introduce in detail the proposed MUD based communication-efficient FL (FedMUD) framework, encompassing the following key states: initialization, broadcasting, local training, uploading and global aggregation. The workflow of this strategy is illustrated in Algorithm 2.

**– Initialization:** The first $M$ rounds of FedMUD serve as the initialization phase, following the conventional FedAvg method. After these rounds, the central server holds an updated global model

---

**Algorithm 2:** Model Update Distillation based FL

---

**Input:** $N$ edge devices with private datasets $\{\mathcal{D}_i\}_{i=1}^{N}$, communication round number $T$, learning rate $\gamma$, initial rounds $M$, local update number $K$, batchsize $B$.

**Output:** FL-trained global model $\omega_g^T$.

**Server Executes:**

    **Initialization:** After $M$ rounds of FedAvg, the central server has $\omega_g^{(M)}$ with $\omega_g^{(M-1)}$, and the device has $\omega_g^{(M-1)}$

    **for** each communication round $t = M, \ldots, T$ **do**

        Obtain $\zeta_g^{(t)}$ by performing model update distillation between $\omega_g^{(t-1)}$ and $\omega_g^{(t)}$ with Eq. (4)

        **for** each device $i = 1, 2, \ldots, N$ **in parallel do**

            Broadcasting $\zeta_g^{(t)}$ to device $i$

            $\zeta_i^{(t)} \leftarrow$ **Device Executes** $(i, \zeta_g^{(t)})$

            Reconstruct the local model $\omega_i^{(t)}$ by one-step gradient descent on $\zeta_i^{(t)}$ with Eq. (8)

        **end**

        $\omega_g^{(t+1)} \leftarrow \frac{1}{N} \sum \omega_i^{(t)}$

    **end**

**end**

**Device Executes** $(i, \zeta_g^{(t)})$**:**

    Reconstruct the global model $\omega_g^{(t)}$ by one-step gradient descent on $\zeta_g^{(t)}$ with Eq. (5)

    Update the local model as $\omega_i^{(t)}$ by local training on $\mathcal{D}_i$ with Eq. (6)

    Obtain $\zeta_i^{(t)}$ by performing model update distillation between $\omega_g^{(t)}$ and $\omega_i^{(t)}$ with Eq. (7)

    Return $\zeta_i^{(t)}$

**end**

---

denoted by $\omega_g^{(M)}$. Subsequent rounds employ model update distillation to reduce the communication load between the central server and edge devices for both downlink (broadcasting the global model) and uplink (uploading local updates) communications.

**– Broadcasting:** In the $t$-th round ($t > M$), the server performs model update distillation between the new global model $\omega_g^{(t)}$ and the global model $\omega_g^{(t-1)}$ in the last round. This process creates a compressed datastream $\zeta_g^{(t)}$, a tensor of length $m_\zeta^{(t)}$, significantly reducing the downlink burden. The optimization solved for this purpose is formulated as:

$$\zeta_g^{(t)} = \underset{\{(x_j, y_j)\}_{j=1}^{m_g^{(t)}}}{\arg\min} \left\| \sum_{j=1}^{m_g^{(t)}} \frac{\partial \ell(\omega_g^{(t-1)}; (x_j, y_j))}{\partial \omega_g^{(t-1)}} - \Delta \omega_g(t-1, t) \right\|_2^2, \tag{4}$$

where $\Delta \omega_g(t-1, t)$ denotes the difference between $\omega_g^{(t)}$ and $\omega_g^{(t-1)}$. Instead of broadcasting the parameter difference, the server broadcasts $\zeta_g^{(t)} = \{(\hat{x}_j, \hat{y}_j)\}_{j=1}^{m}$ to all participating devices.

**– Local Training:** Upon receiving the broadcasted tensor $\zeta_g^{(t)}$, each edge device recovers the intended global model $\omega_g^{(t)}$ using the global model $\omega_g^{(t-1)}$ from the last round. This recovery process is achieved through one-step gradient descent on $\zeta_g^{(t)}$:

$$\omega_g^{(t)} = \omega_g^{(t-1)} - \sum_{(\hat{x}, \hat{y}) \in \zeta_g^{(t)}} \frac{\partial \ell(\omega_g^{(t-1)}; (\hat{x}, \hat{y}))}{\partial \omega_g^{(t-1)}}. \tag{5}$$

After that, for each edge device $i$, started from the model $\omega_g^{(t)}$, it performs local training on its private dataset for $K$ iterations to update its parameters as $\omega_i^{(t,k)}$:

$$\omega_i^{(t,k)} \leftarrow \omega_i^{(t,k-1)} - \gamma \sum_{(x,y) \in \mathcal{B}_i} \frac{\partial \ell(\omega_i^{(t,k-1)}; (x, y))}{\partial \omega_i^{(t,k-1)}} \quad \text{for } k \in [K], \tag{6}$$

where $\omega_i^{(t,0)} = \omega_g^{(t)}$, $\omega_i^{(t)} = \omega_i^{(t,K)}$, and $\mathcal{B}_i$ is the $i$-th random batch drawn from $\mathcal{D}_i$.

Table 2: Wall-clock time and communication cost comparison **under the same test accuracy**, based on the average data uploaded **per device** (MB) and training time (s) after initialization, across four network architectures and datasets including CIFAR-10 and CRCSlides.

| Model | Method | CIFAR-10 (Krizhevsky et al., 2009) | | CRCSlides (Kather et al., 2018; 2019) | |
|---|---|---|---|---|---|
| | | Avg. Data ↑ (MB) | Wall-clock Time (s) | Avg. Data ↑ (MB) | Wall-clock Time (s) |
| GoogLeNet | FedAvg (McMahan et al., 2017) | 764.59 (1×) | 1,555.73 | 578.07 (1×) | 1,113.84 |
| | Top-k (Aji & Heafield, 2017) | 653.50 (1.17×) | 1,509.71 | 458.79 (1.26×) | 959.99 |
| | FedPAQ (Reisizadeh et al., 2020) | 364.21 (2.43×) | 1,045.38 | 284.76 (2.03×) | 502.32 |
| | DAdaQ (Hönig et al., 2022) | 483.63 (1.83×) | 1,159.15 | 450.44 (1.54×) | 955.71 |
| | AdaGQ (Liu et al., 2023) | 471.97 (1.62×) | 1,153.16 | 405.66 (1.71×) | 905.53 |
| | FedMUD | **22.94 (33.33×)** | **965.96** | **22.15 (26.10×)** | **842.22** |
| MobileNet | FedAvg (McMahan et al., 2017) | 707.28 (1×) | 1,605.59 | 547.65 (1×) | 1,346.58 |
| | Top-k (Aji & Heafield, 2017) | 597.82 (1.18×) | 1,662.05 | 402.68 (1.36×) | 1,487.54 |
| | FedPAQ (Reisizadeh et al., 2020) | 341.01 (2.07×) | 1,295.97 | 239.82 (2.28×) | 976.733 |
| | DAdaQ (Hönig et al., 2022) | 497.24 (1.42×) | 1,519.40 | 367.55 (1.49×) | 1,276.83 |
| | AdaGQ (Liu et al., 2023) | 443.49 (1.59×) | 1,373.24 | 285.23 (1.92×) | 1,026.71 |
| | FedMUD | **24.57 (28.79×)** | **1,096.67** | **20.58 (26.61×)** | **854.61** |
| ShuffleNet | FedAvg (McMahan et al., 2017) | 352.64 (1×) | 918.94 | 396.71 (1×) | 1,198.45 |
| | Top-k (Aji & Heafield, 2017) | 320.16 (1.10×) | 1,105.13 | 306.58 (1.29×) | 1,354.72 |
| | FedPAQ (Reisizadeh et al., 2020) | 167.04 (2.11×) | 859.19 | 168.34 (2.36×) | 1,125.46 |
| | DAdaQ (Hönig et al., 2022) | 205.44 (1.72×) | 816.87 | 219.18 (1.81×) | 1,269.67 |
| | AdaGQ (Liu et al., 2023) | 220.26 (1.60×) | 866.52 | 268.05 (1.48×) | 1,394.71 |
| | FedMUD | **18.94 (19.02×)** | **694.17** | **26.75 (14.83×)** | **976.53** |
| ResNet-18 | FedAvg (McMahan et al., 2017) | 1,153.98 (1×) | 2,252.93 | 967.85 (1×) | 1,808.97 |
| | Top-k (Aji & Heafield, 2017) | 848.51 (1.36×) | 1,821.35 | 762.09 (1.27×) | 1,507.63 |
| | FedPAQ (Reisizadeh et al., 2020) | 588.77 (1.96×) | 1,490.32 | 441.94 (2.19×) | 970.22 |
| | DAdaQ (Hönig et al., 2022) | 785.02 (1.47×) | 1,800.66 | 559.45 (1.73×) | 1,159.14 |
| | AdaGQ (Liu et al., 2023) | 682.83 (1.69×) | 1,350.30 | 514.81 (1.88×) | 1,056.11 |
| | FedMUD | **28.16 (40.98×)** | **1,216.71** | **24.38 (38.70×)** | **871.39** |

**– Uploading:** After local training, each device $i$ executes model update distillation between the received global model $\omega_g^{(t)}$ and the updated local model $\omega_i^{(t)}$:

$$\zeta_i^{(t)} = \underset{\{(x_j,y_j)\}_{j=1}^{m_i^t}}{\arg\min} \left\| \sum_{j=1}^{m_g^{(t)}} \frac{\partial \ell(\omega_g^{(t)}; (x_j, y_j))}{\partial \omega_g^{(t)}} - \left( \omega_g^{(t)} - \omega_i^{(t)} \right) \right\|_2^2. \tag{7}$$

Here $m_i^{(t)}$ is related to the difference between $\omega_i^{(t)}$ and $\omega_g^{(t)}$, which varies in each round. Each device $i$ uploads the synthetic samples $\zeta_i^{(t)}$ to the server. The amount of $\zeta_i^{(t)}$ is much smaller than the parameter disparity $\omega_g^{(t)} - \omega_i^{(t)}$, thus the uplink communication burden can be significantly reduced.

**– Global Aggregation:** Upon receiving the uploaded tensor $\zeta_i^{(t)}$, the central server performs one-step gradient descent to recover the intended updated local model $\omega_i^{(t)}$ for each device $i$. This is achieved by addressing the following optimization problem:

$$\omega_i^{(t)} = \omega_g^{(t)} - \sum_{(\hat{x},\hat{y}) \in \zeta_i^{(t)}} \frac{\partial \ell(\omega_g^{(t)}; (\hat{x}, \hat{y}))}{\partial \omega_g^{(t)}}. \tag{8}$$

The server then aggregates the reconstructed local models $\omega_i^{(t)}$ from all selected devices to obtain an updated global model $\omega_g^{(t+1)}$:

$$\omega_g^{(t+1)} = \frac{1}{N} \sum_{i=1}^{N} \omega_i^{(t)}. \tag{9}$$

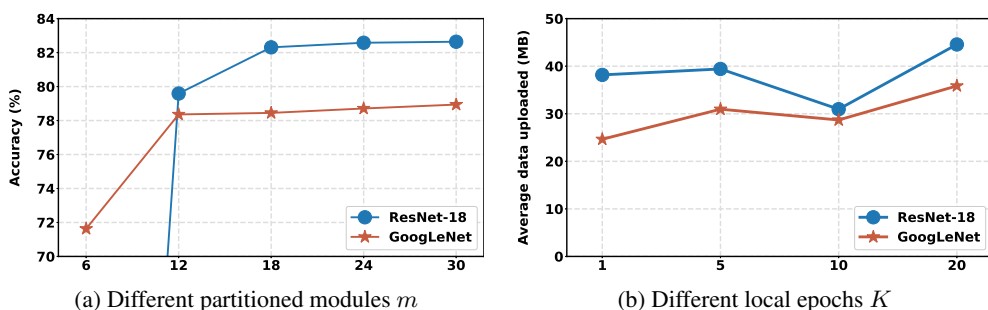

(a) Different partitioned modules $m$       (b) Different local epochs $K$

Figure 2: Ablation analysis of FedMUD's performance with varying (a) number of partitioned modules $m$ and (b) number of local epochs $K$ on the client devices.

## 5 EXPERIMENTS

**Baselines.** We evaluate FedMUD against five state-of-the-art methods, covering quantization, sparsification, and knowledge distillation approaches: 1) Top-k (Aji & Heafield, 2017) is a sparsification technique that reduces communicated gradients by selecting the largest $k$ elements, with $k$ set to 50% of the total parameters. 2) FedPAQ (Reisizadeh et al., 2020) is the first federated learning quantization scheme, which compresses models uploaded by clients and distributed by the server to 8-bit. 3) DAdaQ (Hönig et al., 2022) employs a dual adaptive quantization algorithm that adjusts quantization levels dynamically across rounds and clients. 4) AdaGQ (Liu et al., 2023) adapts quantization levels in each round based on gradient norms and client bandwidth. 5) FedKD (Wu et al., 2022) improves communication efficiency by creating a smaller messenger model via knowledge distillation between the server and clients. While recently published methods like FedDST (Bibikar et al., 2022) and FedCS (Jiang & Borcea, 2023) are relevant, their original papers only test on a three-layer CNN. Thus, we exclude them from our comparison, as their effectiveness on more complex models remains unclear.

**Datasets.** Experiments are conducted on two benchmark datasets: CIFAR-10 and a real-world medical image dataset CRCSlides (Kather et al., 2018; 2019), which is collected for predicting survival from colorectal cancer histology slides. It contains 100,000 images for training and 7,180 images for testing. The image size is $3 \times 32 \times 32$. Similar to prior work (Liu et al., 2023), we use $\sigma_d$ to denote the level of Non-IID data, which corresponds to the fraction of data that belongs to only one class at each device. In our experiments, we set $\sigma_d = 0.2$, representing the scenario where 20% of each local dataset contains samples from only one class, while the remaining 80% contains samples from other classes. Detailed experiment settings are included in the Appendix A.

### 5.1 PERFORMANCE COMPARISON

**Wall-clock Training Time Comparison.** We assess the wall-clock times of our method and comparative methods required to achieve identical accuracy levels. We test four architectures—GoogLeNet, MobileNet, ShuffleNet, and ResNet-18—across two datasets, CIFAR-10 and CRCSlides, and also compare communication costs. The results, shown in Table 2, indicate that on CIFAR-10 using ResNet-18, our method reaches 80% accuracy in 1216.71 s, versus FedAvg's 2252.93 s. Additionally, the average data upload per device for our method is 28.16 MB, a 40.98× reduction compared to FedAvg's 1153.98 MB. On CRCSlides, our method excels further: using GoogLeNet, it achieves 76% accuracy in 842.22 s, while FedAvg takes 1113.84 s. The average data upload per device is 22.15 MB, far lower than FedAvg's 578.07 MB, resulting in a 26.10× reduction. These results highlight our method's suitability for bandwidth-constrained edge devices, improving efficiency while significantly reducing communication costs.

**Impact of the Number of Modules.** To investigate the impact of the number of modules $m$ on our method's performance, we plot the global model's test accuracy on CIFAR-10 by searching over $m \in 6, 12, 18, 24, 30$. The results are shown in Fig. 2a. At $m = 6$, FedMUD struggles to approximate the ResNet-18 model's parameter updates effectively. However, as $m$ increases to 18, accuracy reaches 82.81%. Further increases beyond 18 yield diminishing returns, with accuracy at 82.64% for $m = 30$. For GoogLeNet, an initial accuracy of 78.36% is achieved at $m = 12$, rising slightly to 78.94% at $m = 30$. These findings suggest that with fewer modules, FedMUD struggles

to synthesize tensors that accurately reflect parameter updates, leading to suboptimal performance. As $m$ increases, allowing finer granularity, performance plateaus, with minimal gains from further increases.

**Impact of the Number of Local Epochs.** To explore the influence of local training epochs, denoted as $E$, on the efficacy of our federated learning approach, we conduct experiments to assess the global model's test accuracy on the CIFAR-10 dataset across various $E$ values, specifically $E \in 1, 5, 10, 20, 50$. The results are shown in Fig. 2b. As $E$ increases, our results show a decrease in data uploads and wall-clock time up to a point, indicating improved communication efficiency. However, with $E = 20$ and beyond, while data transfer continues to decrease, computational time rises, revealing a trade-off between reducing communication rounds and increasing computation. This balance is crucial for optimizing federated learning systems under varying network and privacy conditions.

**Model Accuracy Comparison.** We further provide an accuracy comparison under almost the same wall-clock training time. For our method, the total number of communication rounds is set to 100, while the round count for other baseline methods is determined by the rounds reached when their training time exceeds 100 rounds for our method. As shown in Table 3, FedPAQ has the largest total number of rounds among the baselines and also achieved the highest accuracy of 80.36%. FedAvg has the fewest total rounds due to the complete transmission of model information, which severely limits its performance under the bandwidth constraints of edge devices, achieving an accuracy of 80.59%. Under the same time budget, our method allows for the most communication rounds and achieves an accuracy of 82.81%.

Table 3: Accuracy comparison under almost the same wall-clock training time.

| Model | Method | CIFAR-10 | |
|---|---|---|---|
| | | Accuracy (%) | Wall-clock Time (s) |
| GoogLeNet | FedAvg | 77.62 | 1,704.15 |
| | Top-k | 72.17 | 1,685.13 |
| | FedPAQ | 74.35 | 1,675.07 |
| | DAdaQ | 73.57 | 1,686.26 |
| | AdaGQ | 74.09 | 1,668.33 |
| | FedMUD | **78.36** | **1,662.24** |
| ResNet-18 | FedAvg | 80.59 | 2,354.67 |
| | Top-k | 71.64 | 2,347.13 |
| | FedPAQ | 80.36 | 2,344.34 |
| | DAdaQ | 79.78 | 2,359.74 |
| | AdaGQ | 80.22 | 2,355.12 |
| | FedMUD | **82.81** | **2,333.45** |

**Partial Participation with More Devices.** To evaluate the scalability and performance of our method with varying device numbers, we conducted experiments with a fixed device participation rate of 0.2 and total device counts of 50, 100, 200, and 500. The results are shown in Table 4. As expected, increasing the number of devices noticeably affects the model's convergence speed. Specifically, as the number of devices grows, more communication rounds are needed for convergence, likely due to reduced individual device participation per round, which impacts the global model's update frequency. However,

Table 4: Results of 20% participation with more devices under.

| $N$ | Method | CIFAR-10 | |
|---|---|---|---|
| | | Avg. Data ↑ (MB) | Wall-clock Time (s) |
| 50 | FedAvg | 976.52 | 9573.58 |
| | FedMUD | 35.56 (27.46×) | 8756.44 |
| 100 | FedAvg | 1174.91 | 24573.62 |
| | FedMUD | 35.53 (33.06×) | 19538.15 |
| 200 | FedAvg | 1896.43 | 45716.49 |
| | FedMUD | **63.98 (29.64×)** | **40269.37** |

our method demonstrates significant communication efficiency compared to FedAvg. For instance, with 50 devices, our approach cuts data uploaded per device by a factor of 29.64. This advantage is even more pronounced at larger scales: with 500 devices, our method achieves a 42.13-fold reduction in communication overhead compared to FedAvg.

**Without Initialization Phase.** Although in practical industrial applications, the initial model is rarely a randomly initialized one, we intentionally presented results without an initialization phase to rigorously evaluate the performance of our method. As shown in Table 5, even under these conditions, our method still achieves significant reductions in communication overhead. Specifically, our approach requires an average data upload of only 68.84 MB per device, compared to 1768.29 MB for FedAvg. Even when compared to the best-performing quantization-based method, which requires 954.83 MB, our method demonstrates superior communication efficiency.

Table 5: Results of average data uploaded and wall-clock time without initialization phase.

| Model | Method | CIFAR-10 | |
|---|---|---|---|
| | | Avg. Data ↑ (MB) | Wall-clock Time (s) |
| ResNet-18 | FedAvg | 1,768.29 | 4,027.61 |
| | Top-k | 1,498.73 | 4,321.33 |
| | FedPAQ | 954.83 | 3,499.12 |
| | DAdaQ | 1,492.37 | 4,583.69 |
| | AdaGQ | 1,374.82 | 4,267.61 |
| | FedMUD | **132.48 (13.35×)** | **3,733.52** |

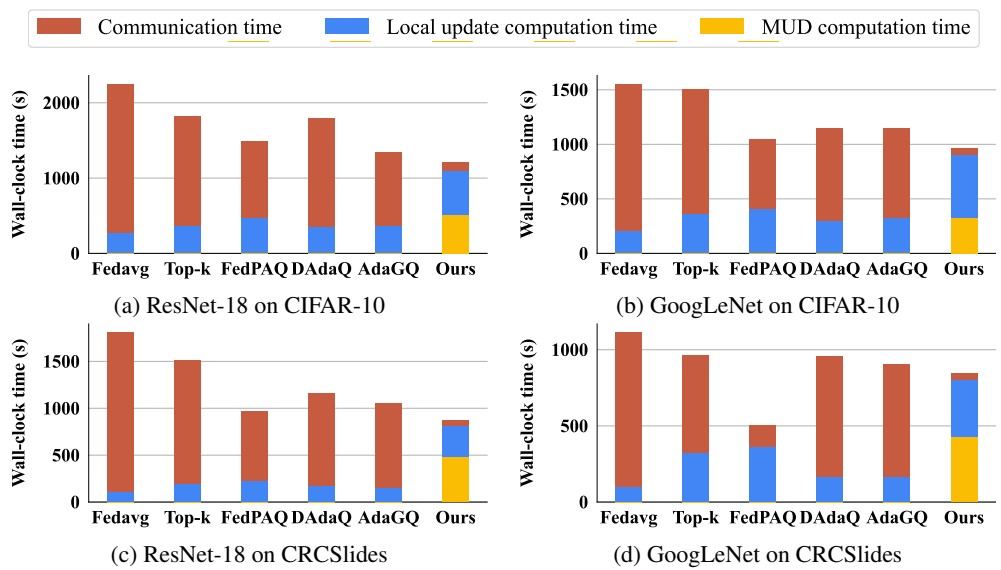

Figure 3: Computation and communication time comparison of FedMUD and baseline methods. We display the proportion of MUD computation in our overall computation time.

## 5.2 COMPUTATION AND COMMUNICATION TIME ANALYSIS

To analyze how FedMUD reduces total training time, we dissect both communication and computation times in Fig.3. For comparison, we also show the proportions of these times for other methods. Our approach requires relatively longer computation time due to the MUD process but benefits from significantly reduced communication time, improving overall training efficiency and device utilization. FedAvg has the shortest computation time, as it transmits complete model information, allowing it to reach target accuracy in fewer rounds. Among quantization-based methods, FedPAQ has the longest computation time, needing more rounds to compensate for information loss from quantization. DAdaQ and AdaGQ use adaptive quantization strategies that reduce communication rounds but require longer communication times. These results highlight the trade-offs between computation and communication times in different federated learning methods, underscoring the effectiveness of FedMUD in optimizing overall training efficiency and device utilization.

## 6 LIMITATIONS DISCUSSION

The MUD procedure requires additional computational cost, and it may be unrealistic to use synthetic tensor to precisely approximate the model parameter update for ultra-large networks. However, in the context of edge computing, models like ResNet-18 are already considered quite large, as most commonly deployed architectures, such as MobileNet and GhostNet, are designed to be more lightweight and efficient.Moreover, the additional computation is offset by the significant reduction in communication overhead. Despite these limitations, we believe our method offers valuable insights and a novel perspective on communication-efficient federated learning.

## 7 CONCLUSION

In this study, we introduced model update distillation-based communication-efficient federated learning (FedMUD), a novel approach where devices and the server synthesize tensor sequences to represent small model updates, rather than transmitting raw model differences. Our key innovation lies in distilling the structural essence of model updates, as opposed to directly compressing them. This enables the transmission of only essential information for synchronization, bypassing the need to transmit the entire set of raw parameter differences. Experimental results demonstrate that FedMUD substantially reduces communication overhead without sacrificing accuracy significantly.

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

## A EXPERIMENT SETTINGS

**Implementations details.** We implement FedMUD alongside baseline methods utilizing the PyTorch framework on a system outfitted with four Nvidia RTX 3090 GPUs. Our simulation involves 10 virtual devices, with a device sampling rate of 1, ensuring each device's participation in model updating during every communication round. The data transmission rate for each device is randomly set between 50 Mbps and 100 Mbps by default. All methods employ stochastic gradient descent for local training, with a learning rate set at 0.01. Specifically, for FedMUD, the LBFGS optimization technique is utilized for synthesizing the tensor sequence, offering the advantage of adaptively controlling the step size to efficiently identify an optimal tensor sequence. The tensor length for each module is set to 10, with the number of iterations also set to 10. For the ResNet-18 architecture, we segmented it into 18 modules, while the GoogLeNet model was partitioned into 11 modules. The batch size is set as 32, and devices are configured to upload model updates at the end of every epoch. All compared methods initialize their networks by pre-training using FedAvg for 30 rounds. Since they share the same initialization process, the experimental results and analyses presented subsequently include only the costs after initialization. It is worth noting that, the initialization step is crucial for achieving training stability. For example, the accuracies of Top-k and AdaGQ drop by as much as 13% and 10% respectively if the initialization step is removed.

## B PYTORCH CODE OF THE CORE IDEA OF MUD

Here we provide the core code of MUD, as shown in Algorithm 3.

---

**Algorithm 3:** PyTorch Code of Model Update Distillation

```python
import torch

def compute_tensor(self, syn_model, n_sample, n_classes, iter_num):
    # Randomly generate m synthetic samples
    syn_size = [n_sample] +
 list(next(iter(self.train_loader))[0].shape[1:])
    syn_inputs = torch.randn(tuple(syn_size), device=self.device,
                requires_grad=True)
    syn_labels = torch.randn((n_sample, n_classes), device=self.device,
                requires_grad=True)
    optimizer = torch.optim.LBFGS([syn_inputs, syn_labels])
    # Get the real gradients
    real_gradients = torch.cat([v.clone().flatten() for v in
 self.dw.values()])
    # Iteratively optimize the synthetic samples
    for iter in range(iter_num):
        def closure():
            optimizer.zero_grad()
            # Get the synthetic gradients
            syn_preds = syn_model(syn_inputs)
            syn_loss = torch.nn.CrossEntropyLoss()(syn_preds,
 syn_labels)
            syn_dw = torch.autograd.grad(loss, syn_model.parameters(),
                    create_graph=True, allow_unused=True)
            syn_gradients = torch.cat([v.flatten() for v in syn_dw])
            loss = torch.nn.MSELoss()(syn_gradients, real_gradients)
            loss.backward()
            return loss
    optimizer.step(closure)
    return syn_inputs, syn_labels
```

---

## C CONVERGENCE ANALYSIS OF FEDMUD

**Analysis for FedMUD.**

**Aussumption 3.1** (Smoothness). All local functions $f_i (i \in [N])$ are $L$-smooth.

**Aussumption 3.2** (Bounded second moment). There exists a constant $\mathbf{G}_{max} > 0$ such that: $\mathbb{E}\left[||\nabla F_i(x)||^2\right] \leq \mathbf{G}_{max}^2, \quad \forall i \in [N], \forall \mathbf{x} \in \mathbb{R}^d$, where $\nabla F_i(x)$ is an unbiased stochastic gradient of $f_i$ at $x$.

**Definition 3.3** (Virtual Sequence). We construct a virtual sequence which is consistent with standard federated learning, directly transmitting parameter update differences each round between server and devices. The global model of this virtual sequence can be represented as:

$$\theta^{(t)} = \omega_g^{(0)} - \sum_{t=1}^{T} \eta \operatorname{avg}_i\left(\gamma \mathbf{G}_{i,v}^{(t-1)}\right), \quad t = 1, ..., T, \tag{10}$$

Where $\left\{\theta^{(t)}\right\}_{t=1}^{T}$ is the global model generated by virtual sequence at round $t$, $\omega_{(g)}^0$ is the initial global model, $\gamma$ is the local learning rate, $\eta$ is the global learning rate, $\mathbf{G}_{i,v}^{(t-1)}$ is the gradient contributions from device $i$ at virtual round $t-1$, and $t$ indexes the global training rounds from 1 to $T$.

The virtual sequence is constructed to establish the relationship between the global sequence $\left\{\omega_g^{(t)}\right\}_{t=1}^{T}$ generated by our method and the global sequence $\left\{\theta^{(t)}\right\}_{t=1}^{T}$ generated by standard federated learning. We will complete the convergence analysis by bounding the distance between $\omega_g^{(t)}$ and $\theta^{(t)}$.

In Sections 3 and 4, for simplicity, we use $\omega_i^{(t)}$ to denote the local model reconstructed by our method after the device receives the sequence tensor. Here we use $\omega_{i,o}^{(t)}$ to represent the original local model. In this section, we mainly analyze how the error between the reconstructed local model $\omega_i^{(t)}$ and the original model $\omega_{i,o}^{(t)}$ affects the convergence.

**Aussumption 3.4** (Bounded error). There exists a constant $\Delta \geq 0$ such that:

$$\mathbb{E}_{\xi \sim \mathcal{S}_i} \left\|\omega_i^{(t)} - \omega_{i,o}^{(t)}\right\| \leq \Delta^2, \quad \forall i \in [N], \forall \mathbf{x} \in \mathbb{R}^d. \tag{11}$$

**Lemma 3.5** (Distance Bound). *For* $\left\{\theta^{(t)}\right\}_{t=1}^{T}$ *of virtual sequence and* $\left\{\omega_g^{(t)}\right\}_{t=1}^{T}, \left\{\omega_i^{(t)}\right\}_{t=1}^{T}$ *of model update distillation FL, we have:*

$$\begin{cases} \mathbb{E}\left[\left\|\theta^{(t)} - \omega_i^{(t)}\right\|^2\right] \leq 4\eta^2(\gamma \mathbf{G}_{\max} + (T-1)\Delta)^2, \\ \\ \mathbb{E}\left[\left\|\theta^{(t)} - \omega_g^{(t)}\right\|^2\right] \leq \eta^2(\gamma \mathbf{G}_{\max} + (T-1)\Delta)^2. \end{cases} \tag{12}$$

**Proof:**

$$\begin{aligned} \mathbb{E}\left[\left\|\theta^{(t)} - \omega_g^{(t)}\right\|^2\right] &= \mathbb{E}\left[\left\|\sum_{\tau=1}^{t} \eta \operatorname{avg}_i\left(\gamma \mathbf{G}_i^{(\tau-1)}\right) - \sum_{\tau=1}^{t} \eta \operatorname{avg}_i\left(\gamma \tilde{\mathbf{G}}_i^{(\tau-1)} + \Delta\right)\right\|^2\right] \\ &\leq \eta^2 \mathbb{E}\left[\left\|\sum_{\tau=1}^{t} \operatorname{avg}_i\left(\gamma \mathbf{G}_i^{(\tau-1)} - (\gamma \tilde{\mathbf{G}}_i^{(\tau-1)} + \Delta)\right)\right\|^2\right] \\ &\leq \eta^2(\gamma \mathbf{G}_{\max} + (T-1)\Delta)^2. \end{aligned} \tag{13}$$

Similarly, $\mathbb{E}\left[\left\|\omega_g^{(t)} - \omega_i^{(t)}\right\|^2\right] \leq eta^2(\gamma \mathbf{G}_{\max} + (T-1)\Delta)^2$. Combining these bounds, we have the following.

$$\mathbb{E}\left[\left\|\theta^{(t)} - \omega_i^{(t)}\right\|^2\right] = \mathbb{E}\left[\left\|\theta^{(t)} - \omega_g^{(t)} + \omega_g^{(t)} - \omega_i^{(t)}\right\|^2\right]$$

$$\leq 2\left(\mathbb{E}\left[\left\|\theta^{(t)} - \omega_g^{(t)}\right\|^2\right] + \mathbb{E}\left[\left\|\omega_g^{(t)} - \omega_i^{(t)}\right\|^2\right]\right) \tag{14}$$

$$\leq 4\eta^2(\gamma\mathbf{G}_{\max} + (T-1)\Delta)^2.$$

**Theorem 3.6** (Convergence Rate). *Let $f_{\max} = f\left(\omega_g^{(0)}\right) - f\left(\omega^\star\right)$, where $\omega^\star$ is the minimizer for $f$, we have:*

$$\frac{1}{T}\sum_{t=1}^{T}\mathbb{E}\left[\left|\nabla f\left(\omega_g^{(t)}\right)\right|^2\right] = O(\frac{f\max}{\eta\gamma T} + \eta\gamma\frac{L\sigma^2}{N} + \eta^2 L^2(\gamma\mathbf{G}_{\max} + (T-1)\Delta)^2). \tag{15}$$

**Proof:** Similar to (Avdiukhin & Kasiviswanathan, 2021) Theorem 2.4.

From smoothness Lipschitz condition on the gradients:

$$\mathbb{E}\left[\left\|\nabla f\left(\omega_g^{(t)}\right) - \nabla f\left(\theta^{(t)}\right)\right\|^2\right] \leq L^2\mathbb{E}\left[\left\|\omega_g^{(t)} - \theta^{(t)}\right\|^2\right] \leq L^2\eta^2(\gamma\mathbf{G}_{\max} + (T-1)\Delta)^2, \text{ and}$$

$$\mathbb{E}\left[\left\|\nabla f_i\left(\omega_i^{(t)}\right) - \nabla f_i\left(\theta^{(t)}\right)\right\|^2\right] \leq L^2\mathbb{E}\left[\left\|\omega_i^{(t)} - \theta^{(t)}\right\|^2\right] \leq 4L^2\eta^2(\gamma\mathbf{G}_{\max} + (T-1)\Delta)^2. \tag{16}$$

First, we bound $\theta^{(t)}$, for $\theta^{(t)}$, we have:

$$\theta^{(t+1)} = \theta^{(t)} - \eta\operatorname{avg}_i\left(\gamma\mathbf{G}_i^{(\tau)}\right). \tag{17}$$

By the smoothness property:

$$\mathbb{E}\left[f\left(\theta^{(t+1)}\right)\right] \leq \mathbb{E}\left[f\left(\theta^{(t)}\right)\right] - \mathbb{E}\left[\left\langle\nabla f\left(\theta^{(t)}\right), \eta\operatorname{avg}_i\left(\gamma\mathbf{G}_i^{(t)}\right)\right\rangle\right] + \frac{L}{2}\mathbb{E}\left[\left\|\eta\operatorname{avg}_i\left(\gamma\mathbf{G}_i^{(t)}\right)\right\|^2\right]. \tag{18}$$

The last term in Eq. 18 can be rewritten as:

$$\frac{L}{2}\mathbb{E}\left[\left\|\eta\operatorname{avg}_i\left(\mathbf{G}_i^{(t)}\right)\right\|^2\right]$$

$$= \frac{\eta^2\gamma^2 L}{2}\mathbb{E}\left[\left\|\operatorname{avg}_i\left(\mathbf{G}_i^{(t)} + \nabla f_i\left(\omega_i^{(t)}\right) - \nabla f_i\left(\omega_i^{(t)}\right)\right)\right\|^2\right]$$

$$= \frac{\eta^2\gamma^2 L}{2}\mathbb{E}\left[\left\|\operatorname{avg}_i\left(\nabla f_i\left(\omega_i^{(t)}\right) + \left(\mathbf{G}_i^{(t)} - \nabla f_i\left(\omega_i^{(t)}\right)\right)\right)\right\|^2\right] \tag{19}$$

$$= \frac{\eta^2\gamma^2 L}{2}\mathbb{E}\left[\left\|\operatorname{avg}_i\left(\nabla f_i\left(\omega_i^{(t)}\right)\right)\right\|^2\right] + \frac{\eta^2\gamma^2 L}{2}\mathbb{E}\left[\left\|\operatorname{avg}_i\left(\mathbf{G}_i^{(t)} - \nabla f_i\left(\omega_i^{(t)}\right)\right)\right\|^2\right]$$

$$= \frac{\eta^2\gamma^2 L}{2}\mathbb{E}\left[\left\|\operatorname{avg}_i\left(\nabla f_i\left(\omega_i^{(t)}\right)\right)\right\|^2\right] + \eta^2\gamma^2\frac{L\sigma^2}{2}.$$

Substituting this into the Eq. 18, get:

$$\mathbb{E}\left[f\left(\theta^{(t+1)}\right)\right] \leq \mathbb{E}\left[f\left(\theta^{(t)}\right)\right] - \mathbb{E}\left[\left\langle\nabla f\left(\theta^{(t)}\right), \eta\operatorname{avg}_i\left(\gamma\nabla f_i\left(\omega_i^{(t)}\right)\right)\right\rangle\right]$$

$$+ \frac{\eta^2\gamma^2 L}{2}\mathbb{E}\left[\left\|\operatorname{avg}_i\left(\nabla f_i\left(\omega_i^{(t)}\right)\right)\right\|^2\right] + \eta^2\gamma^2\frac{L\sigma^2}{2}$$

$$\leq \mathbb{E}\left[f\left(\theta^{(t)}\right)\right] - \eta\mathbb{E}\left[\left\langle\nabla f\left(\theta^{(t)}\right), \operatorname{avg}_i\left(\gamma\nabla f_i\left(\theta^{(t)}\right)\right)\right\rangle\right] \tag{20}$$

$$- \eta\mathbb{E}\left[\left\langle\nabla f\left(\theta^{(t)}\right), \operatorname{avg}_i\left(\nabla f_i\left(\omega_i^{(t)}\right) - \nabla f_i\left(\theta^{(t)}\right)\right)\right\rangle\right]$$

$$+ \frac{\eta^2\gamma^2 L}{2}\mathbb{E}\left[\left\|\operatorname{avg}_i\left(\nabla f_i\left(\omega_i^{(t)}\right)\right)\right\|^2\right] + \eta^2\gamma^2\frac{L\sigma^2}{2}.$$

The first term in Eq. 20 can be simplified by $\mathrm{avg}_i\left(f_i\left(\theta^{(t)}\right)\right)=f\left(\theta^{(t)}\right)$:

$$\eta\mathbb{E}\left[\left\langle\nabla f\left(\theta^{(t)}\right),\mathrm{avg}_i\left(\gamma\nabla f_i\left(\theta^{(t)}\right)\right)\right\rangle\right]=\gamma\eta\mathbb{E}\left[\left\|\nabla f\left(\theta^{(t)}\right)\right\|^2\right]. \tag{21}$$

For the second term in Eq. 20 we have:

$$\begin{aligned}
&\eta\gamma\mathbb{E}\left[\left\langle\nabla f\left(\theta^{(t)}\right),\mathrm{avg}_i\left(\nabla f_i\left(\omega_i^{(t)}\right)-\nabla f_i\left(\theta^{(t)}\right)\right)\right\rangle\right]\\
&\leq\frac{\eta\gamma}{2}\left(\mathbb{E}\left[\left\|\nabla f\left(\theta^{(t)}\right)\right\|^2\right]+\mathbb{E}\left[\left\|\mathrm{avg}_i\left(\nabla f_i\left(\omega_i^{(t)}\right)-\nabla f_i\left(\theta^{(t)}\right)\right)\right\|^2\right]\right)\\
&\leq\frac{\eta\gamma}{2}\left(\mathbb{E}\left[\left\|\nabla f\left(\theta^{(t)}\right)\right\|^2\right]+\mathrm{avg}_i\left(\mathbb{E}\left[\left\|\nabla f_i\left(\omega_i^{(t)}\right)-\nabla f_i\left(\theta^{(t)}\right)\right\|^2\right]\right)\right)\\
&\leq\frac{\eta\gamma}{2}\left(\mathbb{E}\left[\left\|\nabla f\left(\theta^{(t)}\right)\right\|^2\right]+4L^2\eta^2(\gamma\mathbf{G}_{\max}+(T-1)\Delta)^2\right). \quad\text{(According to Eq. 16)}
\end{aligned} \tag{22}$$

For the third term in Eq. 20 we have:

$$\begin{aligned}
&\frac{\eta^2\gamma^2 L}{2}\mathbb{E}\left[\left\|\mathrm{avg}_i\left(\nabla f_i\left(\omega_i^{(t)}\right)\right)\right\|^2\right]\\
&=\frac{\eta^2\gamma^2 L}{2}\mathbb{E}\left[\left\|\mathrm{avg}_i\left(\nabla f_i\left(\theta^{(t)}\right)+\left(\nabla f_i\left(\omega_i^{(t)}\right)-\nabla f_i\left(\theta^{(t)}\right)\right)\right)\right\|^2\right]\\
&\leq\eta^2\gamma^2 L\left(\mathbb{E}\left[\left\|\mathrm{avg}_i\left(\nabla f_i\left(\theta^{(t)}\right)\right)\right\|^2\right]+\mathbb{E}\left[\left\|\mathrm{avg}_i\left(\nabla f_i\left(\omega_i^{(t)}\right)-\nabla f_i\left(\theta^{(t)}\right)\right)\right\|^2\right]\right)\\
&\leq\eta^2\gamma^2 L\left(\mathbb{E}\left[\left\|\nabla f\left(\theta^{(t)}\right)\right\|^2\right]+4L^2\eta^2(\gamma\mathbf{G}_{\max}+(T-1)\Delta)^2\right). \quad\text{(According to Eq. 16)}
\end{aligned} \tag{23}$$

Substituting Eq. 21, 22, 23 into the Eq. 20 and move $\mathbb{E}\left[||\nabla f(\theta^{(t)})||^2\right]$ to the left of the inequality;

$$\mathbb{E}\left[\left\|\nabla f\left(\theta^{(t)}\right)\right\|^2\right]\leq\frac{\left(\mathbb{E}\left[f\left(\theta^{(t)}\right)\right]-\mathbb{E}\left[f\left(\theta^{(t+1)}\right)\right]\right)}{\eta\gamma}+\eta\gamma\frac{L\sigma^2}{2N}+2\eta^3 L^2(\gamma\mathbf{G}_{\max}+(T-1)\Delta)^2. \tag{24}$$

Taking the sum over all iterations:

$$\frac{1}{T}\sum_{t=0}^{T}\mathbb{E}\left[\left\|\nabla f\left(\theta^{(t)}\right)\right\|^2\right]\leq\frac{\left(\mathbb{E}\left[f\left(\theta^{(0)}\right)\right]-\mathbb{E}\left[f\left(\theta^{(T+1)}\right)\right]\right)}{\eta\gamma T}+\eta\gamma\frac{L\sigma^2}{2N}+2\eta^3 L^2(\gamma\mathbf{G}_{\max}+(T-1)\Delta)^2. \tag{25}$$

Finally, we can bound $||\nabla f(\omega_g^{(t)})||$ in terms of $||\nabla f(\theta^{(t)})||$ as:

$$\begin{aligned}
\mathbb{E}\left[\left\|\nabla f\left(\omega_g^{(t)}\right)\right\|^2\right]&\leq 2\left(\mathbb{E}\left[\left\|\nabla f\left(\omega_g^{(t)}\right)-\nabla f\left(\theta^{(t)}\right)\right\|^2\right]+\mathbb{E}\left[\left\|\nabla f\left(\theta^{(t)}\right)\right\|^2\right]\right)\\
&\leq 2\mathbb{E}\left[\left\|\nabla f\left(\theta^{(t)}\right)\right\|^2\right]+2L^2\mathbb{E}\left[\left\|\omega_g^{(t)}-\theta^{(t)}\right\|^2\right]\\
&\leq 2\mathbb{E}\left[\left\|\nabla f\left(\theta^{(t)}\right)\right\|^2\right]+\eta^2 L^2(\gamma\mathbf{G}_{\max}+(T-1)\Delta)^2.
\end{aligned} \tag{26}$$

Substituting this into the inequality above on $\frac{1}{T}\sum_{t=0}^{T}\mathbb{E}\left[\left\|\nabla f\left(\theta^{(t)}\right)\right\|^2\right]$ gives the claimed bound:

$$\frac{1}{T}\sum_{t=0}^{T}\mathbb{E}\left[\left\|\nabla f\left(\omega_g^{(t)}\right)\right\|^2\right]=O\left(\frac{f_{\max}}{\eta\gamma T}+\eta^2 L^2(\gamma\mathbf{G}_{\max}+(T-1)\Delta)^2+\eta\gamma\frac{L\sigma^2}{N}\right). \tag{27}$$

Using the step size $\eta=\sqrt{N}/\sqrt{T}$, we get:

$$\begin{aligned}
&\frac{1}{T}\sum_{t=0}^{T}\mathbb{E}\left[\left\|\nabla f\left(\omega_g^{(t)}\right)\right\|^2\right]\\
&=O\left(\frac{f_{\max}}{\sqrt{NT}\gamma}+\frac{N}{T}L^2(\gamma\mathbf{G}_{\max}+(T-1)\Delta)^2+\gamma\frac{L\sigma^2}{\sqrt{NT}}\right).
\end{aligned} \tag{28}$$

If the approximate error $\Delta$ decays over time $T$ at a rate of $\Delta_t = \frac{\Delta}{T^k}$, where $k > 1$, then our method can achieve the same convergence rate as FedAvg, which is $O(1/\sqrt{NT})$.

