# OpenReview forum: "Communication-Efficient Federated Learning via Model Update Distillation"
_ICLR.cc/2025/Conference — ICLR 2025 Conference Withdrawn Submission_

### Official Review · Reviewer_KMuq · 2024-11-01

**Soundness:** 2
**Presentation:** 2
**Contribution:** 2
**Rating:** 3
**Confidence:** 5

**Summary:**

The paper proposes a new framework to reduce the downlink and uplink communication cost in federated learning. The framework generates synthetic samples both at the server and the client sides at every round, and these samples are used to extract an approximate of the true model update.

**Strengths:**

- The paper is well motivated.

- The idea of using synthetic samples to approximate model updates is original.

**Weaknesses:**

- While using synthetic samples is original, I disagree with the authors about their novelty claim for getting use of the prior knowledge the server has. There are a number of papers exploiting this prior knowledge (or side information) to reduce the communication cost: [1, 2].

- I think the writing could be improved. Especially having an additional section between sections 3.2 and 4 leaves reader in confusion about how (2) and (3) are being used for too long.

- The framework requires generating $m$ synthetic samples both at the server and the clients every round by solving (2). I am not sure if this is a computationally reasonable choice. Also, there is no theoretical guarantee that the generated samples will minimize (2) sufficiently -- leading to an irrecoverable error.

- The main contribution of the paper is sending $m$ synthetic examples instead of model updates in every round to reduce the communication cost. But how do the authors make sure that the size of the synthetic samples is smaller than the size of the model. What is the typical range of $m$ (number of samples)? How is it tuned for different datasets and models?

- It would be good to see a plot that shows the final model performance as $m$ increases.

- "FedPAQ (Reisizadeh et al., 2020) is the first federated learning quantization scheme". I am not sure what the authors meant by this sentence but FedPAQ is definitely not the first quantization scheme for federated learning, e.g., QSGD (2016) and SignSGD (2018). Please correct this in the next revision.

[1] [Adaptive Compression in Federated Learning via Side Information](https://arxiv.org/abs/2306.12625) (AISTATS 2024)

[2] [Wyner-Ziv Estimators for Distributed Mean Estimation with Side Information and Optimization](https://arxiv.org/abs/2011.12160) (AISTATS 2021)

**Questions:**

See weaknesses.

---

### Official Review · Reviewer_c2C7 · 2024-11-03

**Soundness:** 3
**Presentation:** 3
**Contribution:** 3
**Rating:** 6
**Confidence:** 4

**Summary:**

This study proposed FedMUD, a communication-efficient federated learning approach where clients and the server transmit synthesized tensor sequences (synthetic samples) representing model updates. By distilling the core structure of updates instead of compressing raw parameters, FedMUD reduces communication overhead while maintaining accuracy. The effectiveness is supported by empirical results.

**Strengths:**

* The paper is well-written, concise, and easy to follow.
* It takes an interesting approach to studying communication efficiency in FL.
* The empirical results confirm the effectiveness of the proposed method, and the theoretical findings provide insights into understanding it.

**Weaknesses:**

* One of my main concerns is the privacy protection of the proposed FedMUD. The authors mentioned that “MUD's output bears a resemblance to data distillation in terms of its form.” This raises a natural concern: if clients generate synthetic samples and send them to the server, it could potentially reveal the local data distributions on the clients which compromising privacy protection in FL.

* As shown in Figure 3 and discussed in Section 5.2, the proposed method requires significant time for local computation. This raises the question of whether, when using larger models with more trainable parameters, the benefits of employing FedMUD as a communication reduction strategy might become marginal due to the increased local computation time.

* Why did the authors choose LBFGS as the default optimizer for FedMUD? Additionally, the comparison in the experiments might be somewhat unfair if FedMUD uses a different local optimizer than the other baselines. My concern is that the superior empirical results could be partially attributed to the benefits of using the LBFGS optimizer.

**Questions:**

* Some of the notation is unclear. For example, in equation 4, $\zeta_g^{(t)} = \{(x_i, y_i)\}_{j=1}^{m_g^{(t)}}$, but in the following paragraph, it is referred to as $m$ instead of $m_g^{(t)}$. This creates some confusion when trying to understand the number of tensors needed for the global step.

* How is  $m_i^{(t)}$  chosen for the clients?

* I believe the convergence analysis relies significantly on Assumption 3.4, which addresses the discrepancy between the reconstructed model and the original model. In the experiments, what is the magnitude of the constant $\Delta$ ? Is it a minor term or relatively larger compared to other terms?

* How is the wall-clock training time measured? Is it captured using an actual FL system, or are some components estimated through simulation?

* Why refer to the synthetic samples as “tensor sequences”? This seems slightly imprecise—a minor comment.

* Some related paper related to communication efficiency in FL could be discussed:

[1] Hyeon-Woo, N., Ye-Bin, M., and Oh, T. Fedpara: Low-rank hadamard product for communication-efficient federated learning.

[2] Isik, B., Pase, F., Gunduz, D., Weissman, T., and Zorzi, M. Sparse random networks for communication-efficient federated learning.

[3] Wang, Y., Lin, L., and Chen, J. Communication-efficient adaptive federated learning.

---

### Official Review · Reviewer_wsDR · 2024-11-10

**Soundness:** 2
**Presentation:** 2
**Contribution:** 2
**Rating:** 3
**Confidence:** 4

**Summary:**

This paper addresses the communication bottleneck in federated learning (FL) by proposing Model Update Distillation (MUD), a framework that compresses model updates between clients and the server. The authors observe that in standard FL protocols, both clients and server possess identical initial model parameters before local training begins in each round. Despite this shared knowledge, traditional FL methods still transmit complete model parameter differences, leading to redundant communication.

The key technical contribution is a method to represent model updates as synthetic tensor sequences rather than raw parameter differences. After local training, instead of transmitting the full parameter update, each client performs an optimization process to generate a compact tensor sequence. When processed through a single gradient descent step on the initial model, this sequence approximates the intended parameter update while being significantly smaller in size, thus reducing communication overhead.

To address the challenges of handling large neural networks where approximating the entire model update with a single tensor sequence may be impractical, the authors introduce a modular alignment approach. This technique partitions the network into modules, with each module using an independent synthetic dataset to approximate its gradient changes. To manage computational overhead, the number of modules is constrained to remain below the local training batch size.

The authors provide theoretical analysis demonstrating that under certain conditions (when the approximation error decays with time $T$ at a rate of $\Delta_t = \Delta/T^k$ where $k > 1$), their method achieves the same convergence rate as FedAvg ($O(1/\sqrt{NT})$). They also analyze the trade-off between compression and accuracy through empirical evaluation.

The experimental evaluation benchmarks FedMUD against several established baselines including Top-k sparsification, FedPAQ, DAdaQ, and AdaGQ across multiple architectures (GoogLeNet, MobileNet, ShuffleNet, ResNet-18) and datasets (CIFAR-10, CRCSlides). The results show significant reductions in communication overhead (ranging from $14.83\times$ to $40.98\times$) while maintaining comparable model accuracy. The authors supplement these findings with ablation studies examining the impact of partitioned module numbers and local epochs on performance.

**Strengths:**

1. Novel Observation Exploitation: The paper identifies and effectively leverages a previously overlooked aspect of federated learning - the server's prior knowledge of client model parameters at the start of each round. This insight enables a fundamentally different approach to reducing communication overhead.

2. Comprehensive Empirical Validation: The experimental evaluation is methodically structured, testing across multiple architectures (ResNet-18, GoogLeNet, MobileNet, ShuffleNet) and datasets (CIFAR-10, CRCSlides). The results demonstrate substantial communication reduction ($40.98\times$ for ResNet-18) while maintaining model accuracy around 80%.

3. Theoretical Foundation: The paper provides convergence analysis demonstrating that FedMUD can achieve the same asymptotic convergence rate as FedAvg ($O(1/\sqrt{NT})$) under specified conditions on error decay.

4. Implementation Practicality: The method functions as a drop-in replacement for standard Federated Averaging, eliminating the need for modifications to existing network architectures or training protocols.

5. Bilateral Efficiency: The paper comprehensively addresses both uplink and downlink communication efficiency.

**Weaknesses:**

1. Limited theoretical analysis: Although some analysis is present, it has several drawbacks (compared to the analysis of standard FedAvg [1]):

   a) Restrictive theoretical assumptions:
   The paper's convergence analysis relies on a questionable Assumption 3.4 (line 730) about bounded error:

   $\mathbb{E}_{\xi\sim S_i} \| \omega_i^{(t)} - \omega_{i,o}^{(t)} \| \le \Delta^2, \quad \forall i \in [N], \forall x \in \mathbb{R}^d$

   This assumption requires that the error between the reconstructed local model ($\omega_i^{(t)}$) and the original model ($\omega_{i,o}^{(t)}$) is bounded by some constant $\Delta^2$. This is a strong assumption because:
   - $\Delta$ could be arbitrarily large in practice
   - The value of $\Delta$ is unknown before deploying the method
   - There's no practical way to verify or enforce this bound during training
   - The paper doesn't provide empirical evidence that this assumption holds in real scenarios

   b) In line 863, the authors establish a convergence rate $O\left(\frac{f_{\max }}{\sqrt{N T} \gamma}+\frac{N}{T} L^2\left(\gamma \mathbf{G}_{\max }+(T-1) \Delta\right)^2+\gamma \frac{L \sigma^2}{\sqrt{N T}}\right)$ that contains a term $O\left(NT L^2\Delta^2\right)$ not affected by the choice of stepsizes $\eta$ and $\gamma$. This leads to divergence when $T \rightarrow+\infty$ if $\Delta^2$ is fixed. While the authors claim:
   ```
   If the approximate error $\Delta$ decays over time $T$ at a rate of $\Delta_t=\frac{\Delta}{T^k}$, where $k>1$, then our method can achieve the same convergence rate as FedAvg, which is $O(1 / \sqrt{N T})$.
   ```
   The claimed decay of $\Delta$ over time $T$ is not demonstrated in the experiments.

2. Technical documentation issues:
   a) Numerous typos throughout the text:
   - Lines 702, 704, 728: "Aussumption" instead of "Assumption"
   - Line 754: "eta" instead of $\eta$

   b) Confusing notation:
   - Line 730: expectation is taken over some $S_i$ that is never defined

   c) Incomplete mathematical expressions:
   - Line 730: mathematical expression (11) lacks a quantifier $\forall t > 0$

3. Experimental evaluation limitations:
   a) Limited scalability:
   The paper lacks investigation of scalability to larger models (>100M parameters) and larger numbers of clients, which is crucial for real-world federated learning deployments [2,3].

   b) Missing experimental setup details:
   - No information about stepsize parameter tuning for baselines
   - Missing details about batch size B and initial rounds M
   - Only learning rate of 0.01 for FedMUD is specified

   c) Limited baseline comparison:
   Authors claim "We evaluate FedMUD against five state-of-the-art methods" but cite papers from 2017 [4] and 2020 [5] with outdated or unsuitable methods. More relevant state-of-the-art methods can be found in [6].
   Moreover, FedPAQ [5] only compresses uplink communication, making it an unsuitable baseline for this case.
   Additionally, "TopK" typically refers to the compressor [7,8,9] rather than the entire method, making "SGD with TopK" more conventional terminology.

   d) Insufficient evaluation of partial participation:
   While partial participation is fundamental in real-world federated learning systems [2,3], where devices may be unavailable due to various constraints, the paper only presents a basic comparison between FedMUD and FedAvg under this setting.

4. Privacy implications:
   The paper doesn't analyze potential privacy implications of the synthetic tensor sequence, which might leak information about local updates [10].

References:

[1] Khaled et al., "Tighter Theory for Local SGD on Identical and Heterogeneous Data," arXiv 2019

[2] Bonawitz et al., "Towards Federated Learning at Scale: System Design," MLSys 2019

[3] Kairouz et al., "Advances and Open Problems in Federated Learning," 2021

[4] Aji et al., "Sparse communication for distributed gradient descent," arXiv:1704.05021, 2017

[5] Reisizadeh et al., "Fedpaq: A communication-efficient federated learning method with periodic averaging and
quantization," AISTATS 2020

[6] Cheng et al., "Communication-Efficient Distributed Learning with Local Immediate Error Compensation," arXiv 2024

[7] Huang et al., "Lower bounds and nearly optimal algorithms in distributed learning with communication compression," NeurIPS 2022

[8] Stich, "On communication compression for distributed optimization on heterogeneous data," arXiv 2020

[9] Beznosikov et al., "On biased compression for distributed learning," JMLR 2023

[10] Zhu et al., "Deep Leakage from Gradients," NeurIPS 2019

**Questions:**

1. In Table 2, FedAvg's average data upload is comparable to Top-k, which is unexpected since FedAvg should not involve any compression. Could you please clarify this discrepancy?

---

### Official Review · Reviewer_V21P · 2024-11-13

**Soundness:** 2
**Presentation:** 2
**Contribution:** 3
**Rating:** 3
**Confidence:** 5

**Summary:**

The paper studies communication bottleneck of FL

Instead of sending the entire updated parameter set, each client uses an optimization process to produce a compressed tensor sequence that captures only the essential information about the update.

The client generates this tensor sequence by optimizing it to closely approximate the model update. This involves calculating a synthetic gradient (a proxy for the real parameter differences) and adjusting the tensor sequence to minimize the difference between the synthetic and actual updates.

During this optimization, the client uses a Mean Squared Error (MSE) loss function to make the synthetic sequence align as closely as possible with the full parameter change, effectively creating a compressed version of the model update.

**Strengths:**

>
FedMUD introduces a fresh perspective on handling communication bottlenecks by leveraging model update distillation. This approach decouples the updates from the network architecture, which has the potential to transform how updates are transmitted in FL.

>
Cute, well principled idea
+
Some simple but insightful theoretical analysis

**Weaknesses:**

> Very limited experiments - only on cifar10 and CRCSlide

> Try more models ? why just resnet18 and LeNet -- at least a ViT-16 / RN 50 would be more representative beyond toy example

> If we use Error Feedback and w a more light-weight compression operator like top-k we might get better results ?

> EF + Compression as a baseline

> In the exp could you explain the ( comm + Local up dat comm + MUD compute ) times = >
also why is local update comm time different -- isn't it the time for local SGD steps ?

> while you did mention in the limitations -> it would be nice to have a better sense of the costly updates with large param sizes => may be run some experiments w increasing param sizes => compare the update xomputation complexities

**Questions:**

see weakness

---

### Note · Authors · 2024-11-14

I have read and agree with the venue's withdrawal policy on behalf of myself and my co-authors.